# Chronic and Acute Pain and Adverse Economic Outcomes in a 9/11-Exposed Cohort

**DOI:** 10.3390/ijerph21111490

**Published:** 2024-11-09

**Authors:** Jennifer Brite, Junaid Maqsood, Shengchao Yu, Howard E. Alper, James E. Cone

**Affiliations:** 1New York City Department of Health and Mental Hygiene, New York, NY 11101, USA; jmaqsood1@health.nyc.gov (J.M.); syu@health.nyc.gov (S.Y.); halper@health.nyc.gov (H.E.A.); jcone@health.nyc.gov (J.E.C.); 2York College, City University of New York, New York, NY 11451, USA

**Keywords:** chronic pain, acute pain, quality of life

## Abstract

The objective of this study was to determine the association between acute and chronic pain and various economic and quality-of-life outcomes. The study sample was drawn from the World Trade Center Health Registry. Both acute and chronic pain were measured in 2020–2021. Logistic regression models were constructed to determine the odds of several economic and quality-of-life variables: early retirement, low household income, decline in household income, quality of life, and not working due to health. Those who had extreme pain in the last four weeks that interfered with normal work had 3.12 (95% confidence interval (CI): 2.36, 4.39) times the odds of early retirement, 5.34 (95% CI: 3.47, 8.21) times the odds of having a household income below USD 50,000, 2.56 (95% CI: 1.51, 4.33) times the odds of having an income decrease, and 14.4 (95% CI: 11.3, 18.3) times the rate of poor health days compared to those with no pain. Similar results were found for chronic pain. The detrimental effects of pain may influence multiple facets of a patient’s life, and both financial and overall well-being may diminish even several years after a pain diagnosis.

## 1. Introduction

Pain, defined by the International Association for the Study of Pain (IASP) as “an unpleasant sensory and emotional experience associated with, or resembling that associated with, actual or potential tissue damage” [1], is a symptom of several medical conditions or may be of unknown etiology. The experience of acute or chronic pain can have a profound impact on one’s mental, physical, and social well-being. It has been linked to reduced mobility and daily activities [2,3], substance abuse [4,5,6], increased depression [7,8,9] and anxiety [10,11,12], and greater social isolation [13].

The prevalence of pain in the United States (U.S.) is difficult to determine as no objective measure exists and each patient’s experience has a strong subjective component. Pain may be localized to a specific part of the body. For instance, low back pain is a common form of pain, and lifetime prevalence has been estimated to range from 51 to 84% [14,15]. Pain can also be widespread, which is defined as any pain that is bilateral (above and below the waist) [16]. Fibromyalgia, for instance, is a common diagnosis when patients experience widespread pain and is estimated to affect 3% of women and 1% of men [14]. Pain can also be classified as acute or chronic. No formal definition of chronic pain exists but some sources define it as pain beyond the normal time of healing for some health conditions [15,17]; other sources define it as any pain generally lasting more than three months [18,19] or six months [17,20]. A 2018 report from the Centers for Disease Control and Prevention (CDC) estimated that 20.4% of U.S. adults had chronic pain and 8.0% of U.S. adults had high-impact chronic pain (i.e., chronic pain that frequently limits life or work activities) [21], while a National Center for Health Statistics report a year later found similar results [22]. An analysis of almost 50 million Europeans found that 8.85% of respondents experienced daily pain [23], while a study using National Health and Nutrition Examination Survey data found 12.2% of Americans experienced acute pain [24]. Some scholars have posited that chronic pain prevalence may have increased in response to the COVID-19 pandemic, either as part of a post-viral syndrome in those previously infected or due to increased stress in those with already existing pain [25]. The prevalence of acute pain is also difficult to determine because it is rarely captured on population surveys and, by definition, lasts a shorter duration than chronic pain (shorter-duration events are less likely to be captured on a survey than chronic events).

Little is known about the pain experience of trauma survivors in general, and the World Trade Center-exposed cohort specifically. There are several mechanisms that may lead 9/11-exposed persons to experience greater pain. First, approximately 44% of those in collapsed or damaged buildings on 9/11 reported sustaining an injury [26]. Most injuries were not deemed serious or critical, but they included musculoskeletal conditions, fracture and crush injuries, sprains/strains, lacerations, and abrasions [27]. Similarly, many responders may have experienced repetitive strain injuries during the 9/11 clean-up activities [28]. A qualitative study of those injured on 9/11 found many respondents mentioned pain and its effects on their lives [29]. A recent analysis of responders found enrollees had an average pain score of 39.5/100 and that their total anxiety score was positively associated with both pain intensity and pain interference [12]. There is also evidence that pain may increase even among those not directly exposed to a terrorist event. Young et al. examined medical charts in three cities (Baltimore, New York, and Seattle) and found self-reported pain scores statistically significantly increased in each city after 9/11 [30]. However, in a large community-based survey of women in New York and New Jersey before and six months following the World Trade Center attack, no increase in fibromyalgia-like symptoms was found [31]. Although no study, to our knowledge, has examined the association between pain and economic outcomes among disaster-exposed populations, previous analyses of the World Trade Center Health Registry (WTCHR) cohort found that 9/11-related poor health or injury was associated with early retirement in both responders [32,33] and community members [34,35].

A growing body of research has shown that pain affects a person holistically, beyond specific mental and physical outcomes. Specifically, pain can negatively impact a person’s economic outlook and overall quality of life. Previous work on the economic impact of pain has mostly been conducted in the general population and has primarily focused on direct costs, such as clinical expenditures and costs to health agencies or the national economy [36,37,38,39,40]. Moreover, temporality is difficult to establish as many analyses are not longitudinal. For example, several cross-sectional studies have shown that pain patients have lower income and are less likely to be fully employed [24,41,42,43,44]. It is difficult to determine whether those with lower socioeconomic status are more likely to experience pain due to work and other factors, or whether pain leads to a decline in economic outcomes. In one of the few longitudinal analyses of this topic, 50 Dutch chronic refractory complex regional pain syndrome (CRPS) Type 1 patients were shown to have lower household income and were less likely to be employed after their diagnosis [45]. The study of pain and quality of life is difficult as the measurement of quality of life varies from study to study but is often defined as overall physical and mental health. Several studies have found a negative association between pain and quality of life [46,47,48,49,50].

It is important to note that establishing one specific cause for pain is difficult, particularly for events that occurred more than a decade prior. For this reason, we examine pain in this cohort as an exposure that may or may not be directly tied to the events of 9/11. Instead, the goal of this study is to examine the economic effects of pain in a disaster-exposed population using longitudinal data.

The present study will extend previous WTCHR work on this topic with three aims: (1) determine the prevalence of chronic and acute pain in the WTCHR cohort; (2) determine the association between acute pain and various economic and quality-of-life outcomes; (3) determine the association between chronic pain and various economic and quality-of-life outcomes.

## 2. Materials and Methods

### 2.1. Data Collection

The WTCHR began collecting baseline data from 2003 to 2004 for over 71,400 enrollees. The longitudinal cohort is made up of both community members and responders exposed to the events of 9/11/2001. Four follow-up surveys have since been administered: Wave 2 in 2006–2007, Wave 3 in 2011–2012, Wave 4 in 2015–2016, and Wave 5 in 2020–2021. A more detailed description of survey methods has been provided elsewhere [51,52]. The institutional review boards of the CDC and the New York City Department of Health and Mental Hygiene approved the WTCHR protocols.

### 2.2. Analytic Sample

The study sample was restricted to adults on 9/11/2001 who completed Waves 4 and 5. In order to assess early retirement, only those 60 years of age or younger at the time of Wave 5 response were included. To establish temporality, those diagnosed with chronic pain before 2001 or after 2016 were excluded (Figure 1).

### 2.3. Exposures

The Wave 5 survey included two pain measures. Acute pain was measured with an item from the standardized SF-12v2 scale, which was slightly modified for the Wave 5 web survey for consistency with the rest of the survey; these modifications included the removal of question numbers, minor edits to the instrument’s instructions, and removal of a thank you statement after the scale. Specifically, enrollees were asked “During the past 4 weeks, how much did pain interfere with your normal work (including both work outside the home and housework)?” with the following answer choices: Not at all, A little bit, Moderately, Quite a bit, Extremely. The second pain question asked whether enrollees had ever been diagnosed with chronic pain by a doctor or health professional, and if so, the year of the diagnosis.

### 2.4. Outcomes

Several economic and quality-of-life variables were examined: early retirement, low household income, decline in household income, quality of life, and not working due to health. All were measured at Wave 5 unless otherwise noted. Because the analytic sample included only those 60 years of age or younger, all enrollees who self-reported being retired when asked about current employment status were considered to have retired early. Respondents were also able to choose the answer “unable to work due to health” when asked about current work status. Total household income was measured on Waves 4 and 5 with respondents asked to choose one of the following income groups in each wave: Less than USD 25,000, USD 25,000–USD 49,999, USD 50,000–USD 74,999, USD 75,000–USD 99,999, USD 100,000–USD 149,999, or USD 150,000 or more. A binary variable was created for each wave, with those reporting household income below USD 50,000 considered to have low income. Change to reported household income from Wave 4 to Wave 5 was also captured in a new variable with three levels: those who moved to a lower category in Wave 5 were considered to have a decrease in income, those who stayed in the same category were considered to have no change in income, and those who moved to a higher category in Wave 5 were considered to have an increase in income. In order to measure quality of life, enrollees were asked the number of days they experienced poor physical or mental health in the last 30 days.

### 2.5. Other Covariates

All models were adjusted by age on 9/11/2001, gender, race/ethnicity, educational attainment, marital status, injury on 9/11, and post-traumatic stress disorder (PTSD). Gender was categorized as male or female. Race/ethnicity was categorized into five mutually exclusive groups: non-Hispanic White, non-Hispanic Black, Hispanic or Latino, Asian, and Other. Educational attainment at Wave 5 was categorized into four mutually exclusive groups: less than high school, high school diploma only, some college, or bachelor’s degree or higher. Marital status at Wave 5 was categorized as married, widowed, divorced, or never married. Injury on 9/11 was recorded as a binary variable with yes capturing respondents who reported receiving a cut, eye injury, sprain, burn, broken bone, concussion, or other injury on the Wave 1 (baseline) survey; if all Wave 1 injury questions were missing, the binary injury variable created for this analysis was coded as missing. September 11-related PTSD was determined at baseline via a modified version of the PTSD Checklist (PCL). A score of greater than or equal to 44 was considered probable PTSD, while all scores below 44 were classified as no probable PTSD. Those with missing responses to PCL items were classified in the appropriate category if their status could be determined based on completed items; otherwise, probable PTSD status was classified as missing. Prescription pain reliever use in the last 12 months was determined at Wave 5. Finally, community member and responder status were obtained at baseline.

### 2.6. Statistical Analysis

Both pain measures were stratified by each of the economic outcome measures and sociodemographic and other factors. Cochran Armitage tests for trend were used to assess categorical variables, and Mann–Whitney–Wilcoxon and Kruskal–Wallis tests were conducted for age on 9/11/2001 and days of poor mental or physical health due to non-normality. Logistic regression models were constructed to determine the odds of early retirement, low household income, income decrease, and not being employed due to health. All covariates were chosen a priori based on a directed acyclic graph framework (Figure A1). Due to overdispersion, negative binomial models were used to calculate incidence rate ratios for each pain measure and days of poor physical and mental health. In a sensitivity analysis, all models were stratified by responder/community member status and by whether the respondent had ever been prescribed a pain reliever in the last 12 months. We stratified by responder status because uniformed workers have different retirement patterns and options than non-uniformed workers. We also stratified by pain reliever prescription to determine whether pain control would affect the observed results. All missing values were estimated via multiple imputations using the MICE R package 3.15.0. Two-sided *p*-values < 0.05 were considered significant. All analyses were conducted in Posit version 2022.07.1 (formerly RStudio).

## 3. Results

The analytic cohort was predominately male (n = 6542), non-Hispanic White (n = 7740), and married (n = 5934). The average age on 9/11 was 33.4 years. The sample consisted of 5229 responders and 5354 community members. Enrollees’ demographic and social characteristics differed by pain status. Those who did not experience pain were on average younger, better educated, and had higher income. Specifically, those who reported no pain in the last four weeks that interfered with normal work had a mean age of 32.3 (standard deviation (SD): 6.07) on 9/11/2001, 70.4% had at least a bachelor’s degree, and 52.6% were married. Conversely, those with the highest level of pain in the last four weeks that interfered with normal work had a mean age of 35.3 (SD: 5.20) on 9/11/2001, 33.7% had at least a bachelor’s degree, and 59.3% were married (Table 1).

Similar results were seen for chronic pain: those with chronic pain were approximately two years older on 9/11/2001 than those with no chronic pain, 62.1% of those with chronic pain had at least a bachelor’s degree compared to 44.2% of those without chronic pain (Table 2).

Multivariable regression analyses further supported the results observed in bivariate analyses. Those who had extreme pain in the last four weeks that interfered with normal work had 3.12 (95% confidence interval (CI): 2.36, 4.39) times the odds of early retirement, 5.34 (95% CI: 3.47, 8.21) times the odds of having a household income below USD 50,000, and 2.56 (95% CI: 1.51, 4.33) times the odds of having income decrease between Waves 4 and 5 compared to those with no pain. The highest pain group in the last four weeks had 14.4 (95% CI: 11.3, 18.3) times the rate of poor health days compared to those without pain. Similarly, those with chronic pain had about twice the odds of early retirement and having a household income below USD 50,000, 1.61 (95% CI: 1.31, 1.98) times the odds of experiencing an income decrease between Waves 4 and 5, and 8.45 (95% CI: 6.89, 10.4) times the odds of not working due to health compared to those with no chronic pain. Finally, those with chronic pain had 2.8 (95% CI: 2.48, 3.17) times the rate of poor health days compared to those without chronic pain (Table 3).

Sensitivity analyses among responders and community members and also those who had and did not have prescription pain reliever use in the last 12 months produced substantively similar results (Table A1, Table A2, Table A3 and Table A4).

## 4. Discussion

This study found that both acute and chronic pain were associated with diminished economic prospects and quality of life. Specifically, both acute and chronic pain were associated with early retirement, having a household income below USD 50,000, experiencing a decrease in household income between Waves 4 and 5, and greater rates of days of poor health. In addition, acute pain in the last four weeks was associated with a greater likelihood of not working due to health. These results are in keeping with previous work on this topic, which found lower socioeconomic status to be associated with pain [24,41,42,43,44] in cross-sectional analyses. The present analysis also found those in pain had lower quality of life, a finding that has been well-established throughout the scientific literature [46,47,48,49,50].

There are three ways pain may be associated with poor economic outcomes. First, those of lower socioeconomic status may be more likely to experience pain due to poor working conditions and a greater likelihood of doing manual labor, as well as poor access to adequate health care due to fewer financial resources. Second, the relationship may be bidirectional, in which those of lower financial means are more likely to experience pain, but after a pain diagnosis, those across the economic spectrum may see diminished work capability and other financial losses. Finally, pain may be an antecedent to diminished economic prospects. In the case of chronic pain, the present analysis rules out the first possibility by ensuring the pain diagnosis takes place at least four years before the measurement of economic outcomes, thereby minimizing selection effects. For example, income between Waves 4 and 5 was more likely to decrease among those with a chronic pain diagnosis at or before Wave 4. Temporality is more difficult to establish for acute pain which by definition is transitory in nature. Currently, only the Wave 5 survey measured acute pain. Further research is needed to determine if acute pain will predict poorer economic outcomes in future survey waves.

If pain is a determinant of poorer economic outcomes, this relationship may work through several pathways. First, pain may directly affect a person’s ability to work. The WTCHR cohort is made up of a large proportion of professions, such as firefighters, police officers, and other responders, that require physical labor. Pain, particularly chronic pain, may lead to diminished work capacity, such as working shorter hours, and also lead workers in late middle age to retire early [32,33]. Second, physical health conditions may both cause pain and lead to poorer economic prospects. This analysis was adjusted for 9/11-related injuries to disentangle these effects, but confounding by physical health status cannot be ruled out. Next, opioids may also play a mediating role. Although the CDC’s updated guidelines for prescribing opioids recommend nonpharmacologic treatments and nonopioid medications for the treatment of most non-cancer pain [53], as recently as 2015–2018, approximately 6% of U.S. adults reported use of one or more prescription opioids during the past 30 days [54]. Opioid misuse and abuse may lead to lower employment prospects and ultimately lower earnings [55,56]. However, in the current analysis, the association between pain and economic status did not differ by prescription pain reliever use. Finally, pain is associated with increased depression [7,8,9] and anxiety [10,11] and greater social isolation [13], all of which may lead to diminished employment and income.

The present study had several strengths. First, the WTCHR is a large, longitudinal cohort that has been continuously followed for more than 20 years. Next, for chronic pain, temporality was able to be established as respondents provided their year of diagnosis. Finally, WTCHR has a rich set of covariates and demographic information, as well as multiple economic measures, collected across multiple wave surveys.

This analysis had several limitations. First, our sample only included respondents to Waves 4 and 5 surveys; enrollees who were lost to follow-up or who died were excluded. Those in pain or of low socioeconomic status may have been more susceptible to attrition. However, we were able to address this issue by employing multiple imputations in order to analyze the entire baseline sample, so our effect estimates are most likely unaffected. Second, all exposure and outcome measures were self-reported. We cannot rule out the possibility that those who were unemployed or those of lower income may have reported pain status differently than the employed and those of higher income. This may have led to inflated effect estimates. However, it is important to note the SF-12 has been validated in several diverse populations [57] and the chronic pain question asked about a doctor’s diagnosis, not the subjective experience of pain. Additionally, our analysis of acute pain in the last four weeks and each economic outcome was cross-sectional. Similarly, we cannot rule out residual confounding. Finally, Wave 5 was administered from spring 2020 to February 2021. It is possible respondents experienced decreased economic prospects due to external factors such as COVID-19 pandemic-related job loss. However, these economic shocks presumably affected respondents irrespective of pain status, and any possible effect was most likely non-differential, suggesting our effect estimates may underestimate the true effect.

## 5. Conclusions

Pain may produce financial losses at both the macro- and micro-economic levels. The present results suggest the detrimental effects of pain may influence multiple facets of a patient’s life beyond physical discomfort alone. Both financial and overall well-being may diminish even several years after a pain diagnosis. Further research is needed to understand the specific mechanisms that may underlie this association. As the 9/11-exposed cohort reaches retirement age and beyond, designing tailored and effective interventions that address the burden of pain and its effects will become ever more vital to ensure the overall health of this and other disaster-exposed populations.

## Figures and Tables

**Figure 1 ijerph-21-01490-f001:**
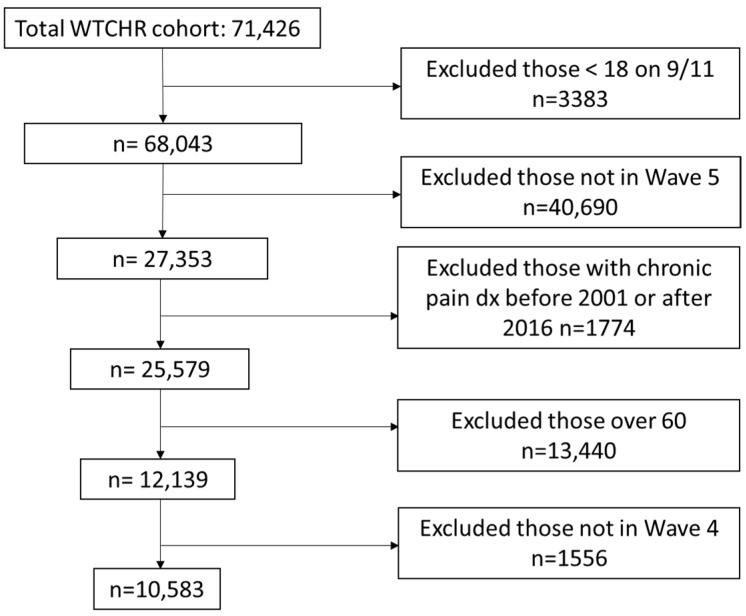
Study selection criteria.

**Table 1 ijerph-21-01490-t001:** Demographic and other characteristics stratified by pain in the last four weeks in the World Trade Center Health Registry.

	Not at All (n = 4405)	A Little Bit (n = 3423)	Moderately (n = 1537)	Quite a Bit (n = 816)	Extremely (n = 258)	*p*-Value
**Age on 9/11**						<0.001 *
Mean (SD)	32.3 (6.07)	33.7 (5.66)	34.5 (5.38)	34.8 (5.00)	35.3 (5.20)	
Median [Min, Max]	33.0 [18.0, 42.0]	35.0 [18.0, 42.0]	36.0 [18.0, 42.0]	36.0 [18.0, 42.0]	36.0 [18.0, 42.0]	
**Gender**						<0.001 *
Male	2529 (57.4%)	2206 (64.4%)	1009 (65.6%)	542 (66.4%)	170 (65.9%)	
Female	1876 (42.6%)	1217 (35.6%)	528 (34.4%)	274 (33.6%)	88 (34.1%)	
**Race/ethnicity**						<0.001 *
Non-Hispanic White	3363 (76.3%)	2546 (74.4%)	1060 (69.0%)	511 (62.6%)	167 (64.7%)	
Non-Hispanic Black	299 (6.8%)	221 (6.5%)	133 (8.7%)	92 (11.3%)	22 (8.5%)	
Hispanic or Latino	394 (8.9%)	377 (11.0%)	224 (14.6%)	157 (19.2%)	49 (19.0%)	
Asian	238 (5.4%)	174 (5.1%)	82 (5.3%)	31 (3.8%)	12 (4.7%)	
Other	111 (2.5%)	105 (3.1%)	38 (2.5%)	25 (3.1%)	8 (3.1%)	
**Educational attainment**						<0.001 *
Less than high school	33 (0.7%)	41 (1.2%)	25 (1.6%)	44 (5.4%)	13 (5.0%)	
High school diploma only	408 (9.3%)	504 (14.7%)	312 (20.3%)	192 (23.5%)	66 (25.6%)	
Some college	848 (19.3%)	879 (25.7%)	465 (30.3%)	262 (32.1%)	91 (35.3%)	
At least a bachelor’s degree	3100 (70.4%)	1985 (58.0%)	727 (47.3%)	316 (38.7%)	87 (33.7%)	
**Marital status**						<0.001 *
Married	2315 (52.6%)	2033 (59.4%)	904 (58.8%)	450 (55.1%)	153 (59.3%)	
Living partner	397 (9.0%)	265 (7.7%)	130 (8.5%)	59 (7.2%)	21 (8.1%)	
Widowed	7 (0.2%)	9 (0.3%)	8 (0.5%)	**	**	
Divorced	167 (3.8%)	176 (5.1%)	83 (5.4%)	72 (8.8%)	20 (7.8%)	
Separated	62 (1.4%)	66 (1.9%)	40 (2.6%)	47 (5.8%)	8 (3.1%)	
Never married	1434 (32.6%)	857 (25.0%)	366 (23.8%)	184 (22.5%)	51 (19.8%)	
**Household income**						<0.001 *
50 k or more	3997 (90.7%)	3053 (89.2%)	1267 (82.4%)	594 (72.8%)	160 (62.0%)	
Less than 50 k	252 (5.7%)	265 (7.7%)	220 (14.3%)	191 (23.4%)	85 (32.9%)	
**Change in income**						<0.001*
Decrease	474 (10.8%)	417 (12.2%)	244 (15.9%)	169 (20.7%)	65 (25.2%)	
Increase	1084 (24.6%)	952 (27.8%)	434 (28.2%)	208 (25.5%)	62 (24.0%)	
Same	2543 (57.7%)	1832 (53.5%)	762 (49.6%)	378 (46.3%)	107 (41.5%)	
**Early retirement**						<0.001 *
No	4014 (91.1%)	2910 (85.0%)	1207 (78.5%)	590 (72.3%)	176 (68.2%)	
Yes	391 (8.9%)	513 (15.0%)	330 (21.5%)	226 (27.7%)	82 (31.8%)	
**Not employed due to health**						<0.001 *
No	4377 (99.4%)	3369 (98.4%)	1439 (93.6%)	611 (74.9%)	143 (55.4%)	
Yes	28 (0.6%)	54 (1.6%)	98 (6.4%)	205 (25.1%)	115 (44.6%)	
**Days of poor health**						<0.001 *
Mean (SD)	1.47 (4.06)	3.61 (6.16)	8.07 (8.37)	15.4 (10.1)	23.0 (8.78)	
Median [Min, Max]	0 [0, 30.0]	1.00 [0, 30.0]	5.00 [0, 30.0]	15.0 [0, 30.0]	27.0 [0, 30.0]	
**Any injury on 9/11**						<0.001 *
No	2979 (67.6%)	1943 (56.8%)	784 (51.0%)	362 (44.4%)	117 (45.3%)	
Yes	1414 (32.1%)	1453 (42.4%)	738 (48.0%)	445 (54.5%)	136 (52.7%)	
**Probable PTSD**						<0.001 *
No	4152 (94.3%)	3019 (88.2%)	1218 (79.2%)	568 (69.6%)	172 (66.7%)	
Yes	209 (4.7%)	373 (10.9%)	294 (19.1%)	234 (28.7%)	83 (32.2%)	

Cell numbers and percentages may not add up to the total due to missing data. * Statistically significant at the *p* = 0.05 level. ** Suppressed due to cell sizes <5.

**Table 2 ijerph-21-01490-t002:** Demographic and other characteristics stratified by chronic pain in the World Trade Center Health Registry.

	Chronic Pain No (n = 8860)	Chronic Pain Yes (n = 1307)	*p*-Value
**Age on 9/11**			<0.001 *
Mean (SD)	33.1 (5.9)	35.0 (5.2)	
Median [Min, Max]	34 [18, 42]	36 [18, 42]	
**Gender**			0.0727
Male	5445 (61.5%)	837 (64.0%)	
Female	3415 (38.5%)	470 (36.0%)	
**Race/ethnicity**			0.4217 *
Non-Hispanic White	6543 (73.8%)	934 (71.5%)	
Non-Hispanic Black	637 (7.2%)	102 (7.8%)	
Hispanic or Latino	971 (11.0%)	188 (14.4%)	
Asian	471 (5.3%)	48 (3.7%)	
Other	238 (2.7%)	35 (2.7%)	
**Educational attainment**			<0.001 *
Less than high school	102 (1.2%)	39 (3.0%)	
High school diploma only	1160 (13.1%)	279 (21.3%)	
Some college	2064 (23.3%)	407 (31.1%)	
At least a Bachelor’s degree	5500 (62.1%)	578 (44.2%)	
**Marital status**			0.1581
Married	4963 (56.0%)	735 (56.2%)	
Living Partner	739 (8.3%)	104 (8.0%)	
Widowed	20 (0.2%)	7 (0.5%)	
Divorced	412 (4.7%)	93 (7.1%)	
Separated	152 (1.7%)	64 (4.9%)	
Never married	2537 (28.6%)	296 (22.6%)	
**Household income**			<0.001 *
50 k or more	7854 (88.6%)	997 (76.3%)	
Less than 50 k	724 (8.2%)	251 (19.2%)	
**Change in income**			<0.001 *
Decrease	1067 (12.0%)	257 (19.7%)	
Increase	2347 (26.5%)	325 (24.9%)	
Same	4868 (54.9%)	620 (47.4%)	
**Early retirement**			<0.001 *
no	7738 (87.3%)	934 (71.5%)	
yes	1122 (12.7%)	373 (28.5%)	
**Not employed due to health**			<0.001 *
no	8662 (97.8%)	1016 (77.7%)	
yes	198 (2.2%)	291 (22.3%)	
**Days of poor health**			<0.001 *
Mean (SD)	3.60 (6.7)	12.2 (10.8)	
Median [Min, Max]	0 [0, 30]	10 [0, 30]	
**Any injury on 9/11**			<0.001 *
no	5412 (61.1%)	628 (48.0%)	
yes	3394 (38.3%)	668 (51.1%)	
**Probable PTSD**			<0.001 *
no	7927 (89.5%)	970 (74.2%)	
yes	837 (9.4%)	319 (24.4%)	

Cell numbers and percentages may not add up to the total due to missing data. * Statistically significant at the *p* = 0.05 level.

**Table 3 ijerph-21-01490-t003:** Regression results of the association between acute and chronic pain and economic outcomes in the World Trade Center Health Registry cohort (n = 10,583).

	Early Retirement			Income Below 50 K			Income Decrease			Days Poor Health			Not Working, Health ^1^		
	OR ^2^	95% CI ^2^	*p*-Value	OR ^2^	95% CI ^2^	*p*-Value	OR ^2^	95% CI ^2^	*p*-Value	IRR ^2^	95% CI ^2^	*p*-Value	OR ^2^	95% CI ^2^	*p*-Value
**Pain in the last 4 weeks**															
Not at all	1.00	Ref.		1.00	Ref.		1.00	Ref.		1.00	Ref.				
A little bit	1.35	1.16, 1.56	<0.001 *	1.22	1.01, 1.47	0.037	1.06	0.91, 1.25	0.4	2.43	2.23, 2.64	<0.001 *			
Moderately	1.89	1.59, 2.25	<0.001 *	2.06	1.55, 2.74	<0.001 *	1.33	1.08, 1.64	0.008 *	5.19	4.46, 6.05	<0.001 *			
Quite a bit	2.69	2.19, 3.31	<0.001 *	3.18	2.18, 4.64	<0.001 *	1.79	1.34, 2.79	<0.001 *	9.62	8.19, 11.3	<0.001 *			
Extremely	3.12	2.36, 4.39	<0.001 *	5.34	3.47, 8.21	<0.001 *	2.56	1.51, 4.33	0.002 *	14.4	11.3, 18.3	<0.001 *			
**Chronic pain**															
Yes	2.19	1.88, 2.55	<0.001 *	2.01	1.63, 2.48	<0.001 *	1.61	1.31, 1.98	<0.001 *	2.8	2.48, 3.17	<0.001 *	8.45	6.89, 10.4	<0.001 *
No	1.00	Ref.		1.00	Ref.		1.00	Ref.		1.00	Ref.		1.00	Ref.	

^1^ Due to sparse cells, odds ratios for days not working due to health could not be estimated for some models. ^2^ OR = Odds Ratio, CI = Confidence Interval, IRR = Incidence Rate Ratio. All models adjusted for marital status, educational attainment, and PTSD status at baseline, any injury on 9/11, race/ethnicity, and age on 9/11/2001. * Statistically significant at the *p* = 0.05 level.

## Data Availability

World Trade Center Health Registry data may be made available following a review of applications to the WTCHR. The data are not publicly available due to privacy or ethical restrictions.

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
