# Peer review of "Chronic and Acute Pain and Adverse Economic Outcomes in a 9/11-Exposed Cohort"

_ijerph, 2024, doi:10.3390/ijerph21111490_

Round 1
Reviewer 1 Report
Comments and Suggestions for Authors
This is an interesting article, but needs some significant improvements, As such:
1. Ho do the authors differentiate between pain due to WTC event inflicted injuries and somatization? How do the authors address this difference in their analyses? Need to explain.
2. Have the authors considered the jobs that offer early retirement due to the high risk of the job, or some federal jobs? These early retirements have nothing to do with the WTC 9/11 event, but with the nature of the actual job. The authors need to address this aspect in their analyses.
3. Looking at the "other covariates" included in the analyses. are marital status and educational attainment proven risk factors for current pain level?? All of the "Other covariates" included in the analyses need to satisfy the definition of a confounder. As such, the authors need to include the corresponding references for EACH independent variable included in the analyses proving that this variable/confounder is a risk factors for outcome, AND is a risk factor for the exposure, or associated with the exposure, but not the result of the exposure. Otherwise, all the analyses will need to be redone with independent variables that satisfy the definition of a confounder, to avoid having unreliable analysis results.
4. Have the authors applied the Directed Acyclic Graph (DAG) theory to determine the minimum sufficient set of confounders to be included in their analyses? These details need to be included in the paper.
Reviewer 2 Report
Comments and Suggestions for Authors
This is a well conducted study which aims to estimate the associations between acute and chronic pain and various economic and quality of life outcomes among participants in the New York City World Trade Center Health Registry (WTC-HR).
Introduction:
· Consider expanding on the statement starting on line 49 “…prevalence of acute pain is also difficult to determine because …” to explicitly state that shorter-duration events are less likely to be captured on a survey than chronic events.
· The sentence starting on line 50 “An analysis of almost 50 million Europeans…” seems a non sequitur and should be moved further up in the introduction.
· The term “In addition” is overused throughout the manuscript.
Methods:
· The level of detail in methods section is very good including the description of alternations made to the standardized instrument.
· Models specification is also well described but some of the factors adjusted for, specifically injury on 9/11 and PTSD, may be along the causal pathway rather than confounders, a DAG would help clarify the variables selected.
· The two sentences beginning on line 170 with “Prescription pain reliever 170 use in the last 12 months …” Belong in the previous paragraph where covariates are described.
Results:
· Beginning on line 179 the authors describe demographic differences in pain prevalence as “more likely to be…”. This should be described as average differences (e.g., on average were older…) rather than as probabilities.
· Table 1: Is the p-value on Table 1 reporting results from a test for trend or any difference? A test for trend would be more informative.
· Table 2: Consider switching the order of the columns so they are directionally like Table 1 (‘no pain’ to the left of ‘pai’n)
Discussion
In the limitations section the authors should add how these limitations may have impacted the study findings.
